# The role of cofeeding arthropods in the transmission of *Rickettsia felis*

**Chanida Fongsaran**[1¤]**, Krit Jirakanwisal**[1,2]**, Natthida Tongluan**[1,2]**, Allison Latour**[1]**, Sean Healy**[1]**, Rebecca C. Christofferson**[1]**, Kevin R. Macaluso**[1,2]*

1 Vector-Borne Disease Laboratories, Department of Pathobiological Sciences, School of Veterinary Medicine, Louisiana State University, Baton Rouge, Louisiana, United States of America, 2 Department of Microbiology and Immunology, College of Medicine, University of South Alabama, Mobile, Alabama, United States of America

¤ Current address: Department of Pathology, the University of Texas Medical Branch, Galveston, Texas, United States of America

* kmacaluso@southalabama.edu

**Data Availability Statement:** All relevant data are within the manuscript.

**Funding:** This work was supported by the National Institutes of Health [AI122672 and AI077784 to

## Abstract

*Rickettsia felis* is an emerging etiological agent of rickettsioses worldwide. The cosmopolitan cat flea (*Ctenocephalides felis*) is the primary vector of *R. felis*, but *R. felis* has also been reported in other species of hematophagous arthropods including ticks and mosquitoes. Canines can serve as a bacteremic host to infect fleas under laboratory conditions, yet isolation of *R. felis* from the blood of a vertebrate host in nature has not been realized. Cofeeding transmission is an efficient mechanism for transmitting rickettsiae between infected and uninfected fleas; however, the mechanism of transmission among different orders and classes of arthropods is not known. The potential for *R. felis* transmission between infected fleas and tick (*Dermacentor variabilis*) and mosquito (*Anopheles quadrimaculatus*) hosts was examined via cofeeding bioassays. Donor cat fleas infected with *R. felis* transmitted the agent to naïve *D. variabilis* nymphs via cofeeding on a rat host. Subsequent transstadial transmission of *R. felis* from the engorged nymphs to the adult ticks was observed with reduced prevalence in adult ticks. Using an artificial host system, *An. quadrimaculatus* exposed to a *R. felis*-infected blood meal acquired rickettsiae and maintained infection over 12 days post-exposure (dpe). Similar to ticks, mosquitoes were able to acquire *R. felis* while cofeeding with infected cat fleas on rats infection persisting in the mosquito for up to 3 dpe. The results indicate *R. felis*-infected cat fleas can transmit rickettsiae to both ticks and mosquitoes via cofeeding on a vertebrate host, thus providing a potential avenue for the diversity of *R. felis*-infected arthropods in nature.

## Author summary

Primarily associated with the common cat flea, *Rickettsia felis* is an intracellular bacterial pathogen that can be transmitted from the flea to vertebrate hosts. This flea-borne infection has now been identified worldwide as a human pathogen. In addition to fleas, other blood feeding arthropods including ticks and mosquitoes are being recognized as possible vectors of *R. felis*. Although the mammalian infectious source for arthropods is still

KRM]. The funders had no role in study design, data collection and analysis, decision to publish, or preparation of the manuscript.

**Competing interests:** The authors have declared that no competing interests exist.

unknown, cofeeding transmission of *Rickettsia* is known to occur between vectors of the same species. However, potential for flea transmission of *R. felis* to other orders and classes of arthropods is unknown. Here, we examined the potential for fleas to transmit *R. felis* to American dog ticks and mosquitoes during feeding events on rat hosts. Our data suggested that ticks and mosquitoes can be infected when simultaneously feeding on a host with *R. felis*-infected cat fleas.

## Introduction

*Rickettsia felis* is an emerging pathogen typically associated with clinical symptoms similar to other rickettsioses including headache, chills, fever, myalgia and rash followed by complications involving neurological systems [1]. The first human infection with *R. felis* was described nearly three decades ago in the United States [2]; subsequently, case reports have increased throughout Central and South America, sub-Saharan Africa, Europe, and Asia over the last 25 years [3]. The cat flea, *Ctenocephalides felis*, functions as a primary reservoir host and vector, transmitting the agent both vertically and horizontally [4]. Transovarial transmission directly from female to progeny has been reported in laboratory-reared *R. felis*-infected cat flea colonies but with high variability in transmission rates, suggesting that vertical transmission alone is likely inadequate to maintain *R. felis* within flea populations [5–7]. Thus, horizontal transmission of *R. felis* from infected to naïve fleas via cofeeding was determined to be sufficient to perpetuate the agent within populations of cat fleas [8,9]. Given the success of cofeeding as a means of flea-to-flea transmission, it is plausible that cofeeding may also be an effective transmission strategy between infected fleas and naïve non-flea, hematophagous arthropods.

Nucleic acid from *R. felis* and related *R. felis*-like organisms (RFLOs) has been detected in other hematophagous arthropods, including more than 40 species of fleas, ticks, and mosquitoes [4,10]. Field surveys have identified *R. felis* in wild-caught ticks [11–13] and laboratory studies have demonstrated *R. felis* was vertically transmitted by the American dog tick, *Dermacentor variabilis*, when adult ticks were exposed via capillary feeding [14]. More recently, in sub-Saharan African countries where *R. felis* has been reported as an emerging cause of unknown fever, *R. felis* has been detected in mosquitoes [3,15,16]. Detection of *R. felis* in male mosquitoes in China and unfed adult mosquitoes in the United States suggests vertical transmission in nature [17,18]. Furthermore, horizontal transmission of *R. felis* by *Anopheles gambiae* was reported using a laboratory model of infection [19]. Thus, while field and laboratory studies support the transmission potential of *R. felis* by alternate vectors, conduits for introduction of the agent into arthropod populations remain undefined.

Cat fleas have a broad host range, infesting feral hosts as well as commonly being identified on companion animals [20]. Within domestic or peri-domestic environments, vertebrate hosts are exposed to fleas, mosquitoes, and ticks. Thus, concurrent feeding by infected fleas and other arthropods on the same host could contribute to transmission of pathogens. Indeed, transmission of *R. felis* from infected cat fleas to naïve rat fleas (*Xenopsylla cheopis*) cofeeding on vertebrate hosts was demonstrated [9]; however, potential for flea transmission of *R. felis* to other orders and classes of arthropods is not known. To test the hypothesis that *R. felis*-infected cat fleas may contribute to dissemination of the agent to other hematophagous vectors, cofeeding bioassays were employed to examine rickettsial transmission from infected cat fleas (*C. felis*) to ticks (*D. variabilis*) and mosquitoes (*Anopheles quadrimaculatus*) on a vertebrate host. The data suggest *R. felis*-infected cat fleas transmit rickettsiae to other hematophagous vectors via cofeeding on a vertebrate host.

## Materials and methods

### Ethics statement

All animals use in this research was performed under the approval of the Louisiana State University (LSU) Division of Laboratory and Animal Medicine (DLAM) with LSU IACUC 17–115.

### Rickettsial propagation

*Rickettsia felis* str. LSU, originally isolated from cat fleas, was propagated in the *Ixodes scapularis*-derived embryonic cell line (ISE6), which was maintained in modified L15B growth medium as previously described [21]. Rickettsiae were semi-purified from ISE6 cells and enumerated by the LIVE/DEAD *Bac*Light viability stain kit (Invitrogen, Eugene, OR), as described previously [22]. To generate infected bloodmeals, concentrations of *R. felis* (passage 3) were adjusted to $3 \times 10^{10}$ rickettsiae in 700 μl (cat fleas) or 2 ml (mosquitoes) of heat-inactivated (HI) bovine blood.

### Fleas, ticks, and mosquitoes

Newly emerged cat fleas were obtained from Elward II (Soquel, CA) and maintained on defibrinated bovine blood (HemoStat Laboratories, Dixon, CA) within an artificial dog feeding unit [23]. Prior to use in bioassays, genomic DNA (gDNA) was extracted from a portion of fleas and confirmed *Rickettsia*-negative by quantitative real-time PCR (qPCR) [24,25]. In order to generate *R. felis*-infected donor cat fleas, newly emerged cat fleas were pre-fed HI bovine blood for 24 hrs, starved for 5–6 hrs, then exposed to the *R. felis*-infected bloodmeal ($3 \times 10^{10}$ rickettsiae/700 μl) for 24 hrs [8]. *Rickettsia*-exposed cat fleas were fed on an uninfected bloodmeal for 48 hrs, and 20 fleas of random sex were collected to assess *R. felis* prevalence in the donor population by qPCR. For cofeeding bioassays, cat fleas treated in the same manner, except being exposed to a bacteria-free bloodmeal for the 24 hrs exposure, served as the control, uninfected 'donor' fleas.

*Rickettsia*-free *D. variabilis* (originally provided by Dr. Daniel Sonenshine) were maintained in a controlled environmental chamber at 27˚C, with 92% relative humidity, and a 12:12 (light:dark) cycle [26,27].

*Anopheles quadrimaculatus* were provided by Dustin Miller (Centers for Disease Control and Prevention for distribution by BEI Resources, NIAID, NIH) and maintained in a controlled environmental chamber at 27˚C, with 92% relative humidity, and a 12:12 (light:dark) cycle. Adults were fed with a 10% sucrose solution [28].

### Flea to tick cofeeding transmission bioassay

To assess acquisition of *R. felis* by *D. variabilis* cofeeding with *R. felis*-infected cat fleas, 30 nymphal ticks were encapsulated on Sprague Dawley (SD) rats in the top portion of a 50 ml conical tube and allowed to attach and feed as described [14]. After 2 days, unattached ticks were removed from the feeding capsule and *R. felis*-infected donor cat fleas (10 male/25 female) were placed into the same feeding capsule with *D. variabilis* experimental group. The control group of ticks were cofed with 20 (10 male/10 female) *Rickettsia*-free cat fleas. After 24 hrs of cofeeding, 10 nymphal ticks were collected from each rat and subjected to gDNA extraction. Likewise, a portion of the donor cat fleas were subsampled for gDNA extraction. The remaining nymphs were left in the feeding capsule and the remaining viable cat fleas were added back to the capsule daily until tick engorgement (~5 days). After natural detachment, a portion of ticks and cat fleas were used for gDNA extraction, with the remaining ticks being

allowed to molt to adult before assessment of rickettsial infection. Each experiment was conducted twice independently.

## Mosquito exposure to *R. felis* by membrane feeding assay

To assess the ability of *An. quadrimaculatus* to harbor *R. felis* at various time points after exposure, female mosquitoes were starved for 5 hrs, before being allowed to feed on a glass membrane feeder containing either a *R. felis*-infected HI bovine blood meal ($3 \times 10^{10}$ rickettsiae/2 ml) or an uninfected blood meal. Mosquitoes fed through a Parafilm membrane and across a glass feeding chamber containing recirculating warm water to keep the blood at 37° C [29]. After the blood meal, engorged female mosquitoes were transferred to a new cage and maintained on a 10% sucrose feeding solution in the chamber. Mosquitoes were sampled and individually processed at 1, 3, 5, 7, 9, and 12 day post-exposure (dpe). Each experiment was conducted twice independently.

## Flea to mosquito cofeeding transmission bioassay

To assess acquisition of *R. felis* by *An. quadrimaculatus* cofeeding with *R. felis*-infected cat fleas, *R. felis*-infected donor cat fleas (10 male/25 female) were encapsulated on SD rats in the top portion of a 50 ml conical tube and allowed to feed for 24 hrs. Subsequently, 10 uninfected mosquitoes per day were placed into the same feeding capsule and allowed to cofeed with fleas for 20 mins once per day for 3 days. Fed female mosquitoes were collected and transferred to new cage and provided access to a cotton ball soaked in 10% sucrose for 24 hrs before being assessed for rickettsial infection. Ten female *R. felis*-infected donor cat fleas were collected daily for immediate gDNA extraction. On separate vertebrate hosts, groups of *Rickettsia*-free donor fleas were cofed with mosquitoes in the same manner and served as the negative infection control. Each experiment was conducted twice independently.

## Sample preparation and rickettsial quantification by PCR

All flea, tick, and mosquito samples were washed with 10% bleach for 5 mins, 70% ethanol for 5 mins, and three times with sterile distilled $H_2O$ for 5 mins to remove environmental DNA. Each sample was transferred to a 1.7 ml microcentrifuge tube and ground with sterile plastic pestles in liquid nitrogen. Extraction of gDNA was performed using the DNeasy Blood and Tissue Kit (Qiagen, Germantown, MD) following the manufacturer's instructions with a final elution in 35 μl of UltraPure DNAse/RNAse free distilled $H_2O$ (Invitrogen, Grand Island, NY). A negative environmental control, a 1.7 ml microcentrifuge tube containing buffer(s) but no sample, was included in each DNA extraction procedure as a negative environmental control for qPCR.

Assessment of rickettsial infection by qPCR was performed with a LightCycler 480 Real-Time PCR system (Roche, Indianapolis, IN). The plasmid pCR4-TOPO-RfelB was used as a standard template to create serial 10-fold dilutions and matched with RfelB primers and probe as previously described [24,25]. For *An. quadrimaculatus* gene expression measurements, mitochondrial DNA (mtDNA) and *ribosomal protein S7 gene* were used as reference gene [30,31].

## Statistical analyses

All statistical analyses were performed in R Studio (version 1.3.1093) with base R (version 3.6.3). To compare concentration of *R. felis* in each arthropod, the concentration of each sample from three technical replicates was averaged. Two experimental trials (biological replicates)

were conducted and data were aggregated over both trials after determination of no significant difference between trials. The rickettsial load was compared using the Kruskal-Wallis non-parametric analysis of variance (*kruskal.test*). When warranted, post-hoc test for pairwise differences was performed using the Dunn Test (*dunnTest*, package FSA). To test for differences in proportions of infected arthropods, the *prop.test* function was performed, where appropriate. In all analyses, statistical significance was assessed at the 95% confidence level.

## Results

### Transmission of *R. felis* from fleas to cofeeding ticks

To determine the potential for *R. felis* transmission from infected cat fleas to other arthropods, we first examined cofeeding transmission from cat fleas to *D. variabilis* ticks on a vertebrate host, SD rats. Infection status of the donor cat fleas was assessed following the experimental infection and determined to be 100% (23/23) infected with *R. felis*. *Rickettsia*-free nymphal *D. variabilis* actively feeding on a vertebrate host were cofed with either *R. felis*-infected cat fleas or uninfected cat fleas as a control and transmission was assessed in individual ticks that cofed for 24 hrs, at tick engorgement (4–5 days), and after molting to the adult stage. *R. felis* was detected in 75% (15/20) *D. variabilis* nymphs that were forcibly removed from the host 24 hrs after cofeeding. Comparably, 81.2% (13/16) of 4–5 days fed, engorged ticks were positive for *R. felis*; and, after molting, 56.2% (9/16) of adult ticks were positive. None of the ticks that cofed with uninfected fleas were positive for *R. felis* (Table 1). While a reduction in the overall proportion of adult ticks that were *R. felis*-positive compared to nymphal stages was observed, there were not statistically significant differences in the proportions of infected ticks among cofeeding duration and life cycle stages of exposed ticks. Further, there was no statistical difference in the rickettsial load from ticks among life cycle stages. These observations demonstrate that transmission of *R. felis* from infected cat fleas to immature ticks via cofeeding allows for a mechanism to generate *R. felis*-infected adult ticks.

### Susceptibility of mosquitoes to *R. felis* infection

To assess the susceptibility of *An. quadrimaculatus* to acquire *R. felis* from oral exposure, individual mosquitoes were offered an *R. felis*-infected bloodmeal and infection was assessed at 1, 3, 5, 7, 9, and 12 dpe. At each time point, 100% of blood fed mosquitoes had detectable *R. felis* in whole body samples, compared to 0% of the control (mock-exposed) individuals. Due to the lack of variance in either group, statistical analysis could not be performed. Mosquitoes tested

**Table 1. Detection of rickettsiae by qPCR in *D. variabilis* nymphs cofed with uninfected or *R. felis*-infected cat fleas for 24 hours on a vertebrate host.** Rickettsial infection was also assessed in ticks after completion of blood meal acquisition (4–5 days) and as newly molted adults.

| Life Stage | Cat fleas cofed with | % Positive (n) | Median Log10 Concentration (Range) |
|---|---|---|---|
| Nymph | | | |
| (24 hours fed) | *R. felis*-infected | 75% (20) | 1.72 (ND, 3.66) |
| | Uninfected | 0% (20) | ND |
| (4–5 days fed) | *R. felis*-infected | 81.2% (16) | 1.37 (ND, 3.06) |
| | Uninfected | 0% (10) | ND |
| Adults (post-molt) | | | |
| | *R. felis*-infected | 56.2% (16) | 1.10 (ND, 2.71) |
| | Uninfected | 0% (10) | ND |

ND, not detected

at day 1 post-exposure had the highest median rickettsial load ($5.54 \times 10^5$) per mosquito, which decreased significantly by day 3 post-exposure ($3.57 \times 10^2$). The significant changes when comparing rickettsial loads 1 dpe to the later time points is likely an indication that the *R. felis* detected was remnants of the bolus in the offered blood meal. After day 3, the rickettsial load in mosquitoes increased at days 5 ($2.21 \times 10^4$), 7 ($2.36 \times 10^4$), 9 ($4.04 \times 10^4$), and 12 ($1.44 \times 10^4$) post-exposure (Fig 1). Days 5–12 were not statistically significant from one another, while day 12 was not statistically different from any time points, likely due to the small sample size at that day (n = 7).

## Transmission of *R. felis* from fleas to cofeeding mosquitoes

Naïve *An. quadrimaculatus* were able to acquire infection through cofeeding with *R. felis*-infected cat fleas. Donor *R. felis*-infected cat fleas were assessed for infection on the same days of cofeeding as *An. quadrimaculatus*. At days 1, 2, and 3 post-cofeeding, 60% (12/20), 65% (13/20), and 70% (14/20), respectively, of cat fleas were positive for *R. felis*. There was no significant difference in the proportions of *R. felis*-infected cat fleas in cofed populations among the time points assessed. Of the cofeeding *An. quadrimaculatus*, 31.6% (6/19), 40% (8/20), and 60% (12/20) were *R. felis*-positive on days 1, 2, and 3 post-exposure, respectively. Similar to cat

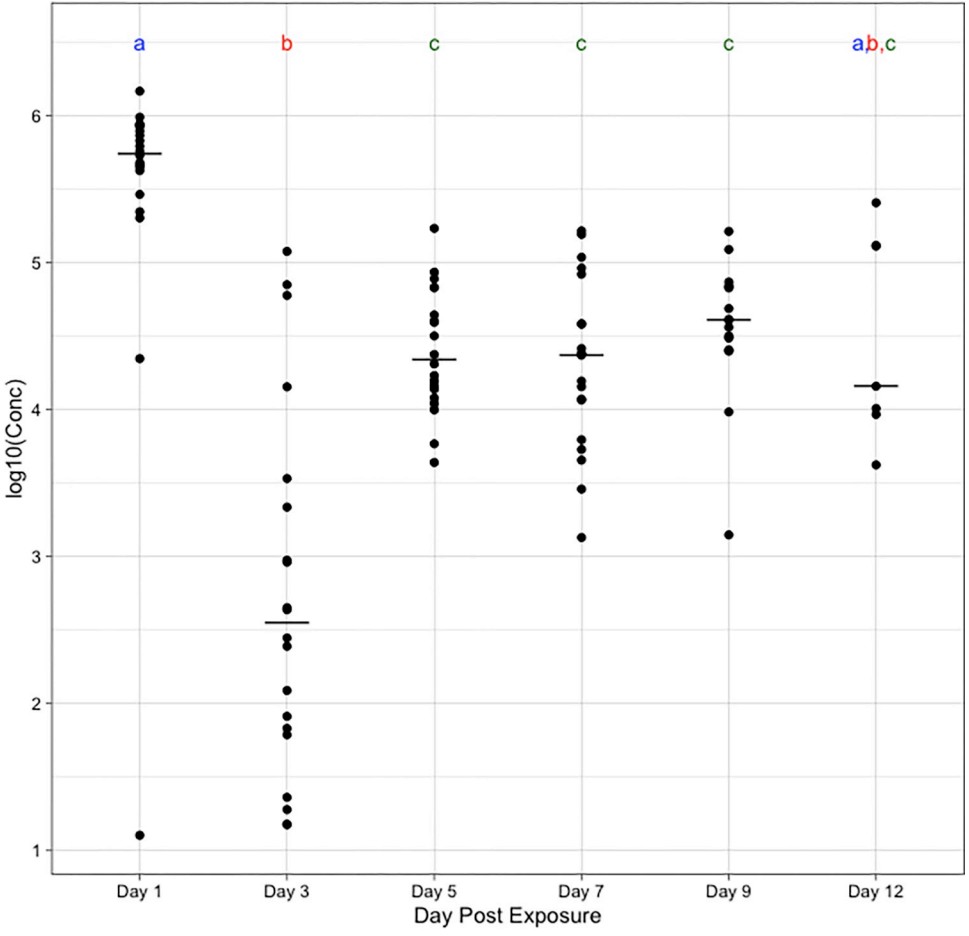

**Fig 1. Concentration (log10) of rickettsiae in individual whole mosquitoes per time point after exposure to an *R. felis*-infected bloodmeal.** Mosquitoes ingested *R. felis* and maintained infection over the course of 12 days. Lines represent the median and letters indicate significance grouping according to Dunn's post-hoc test.

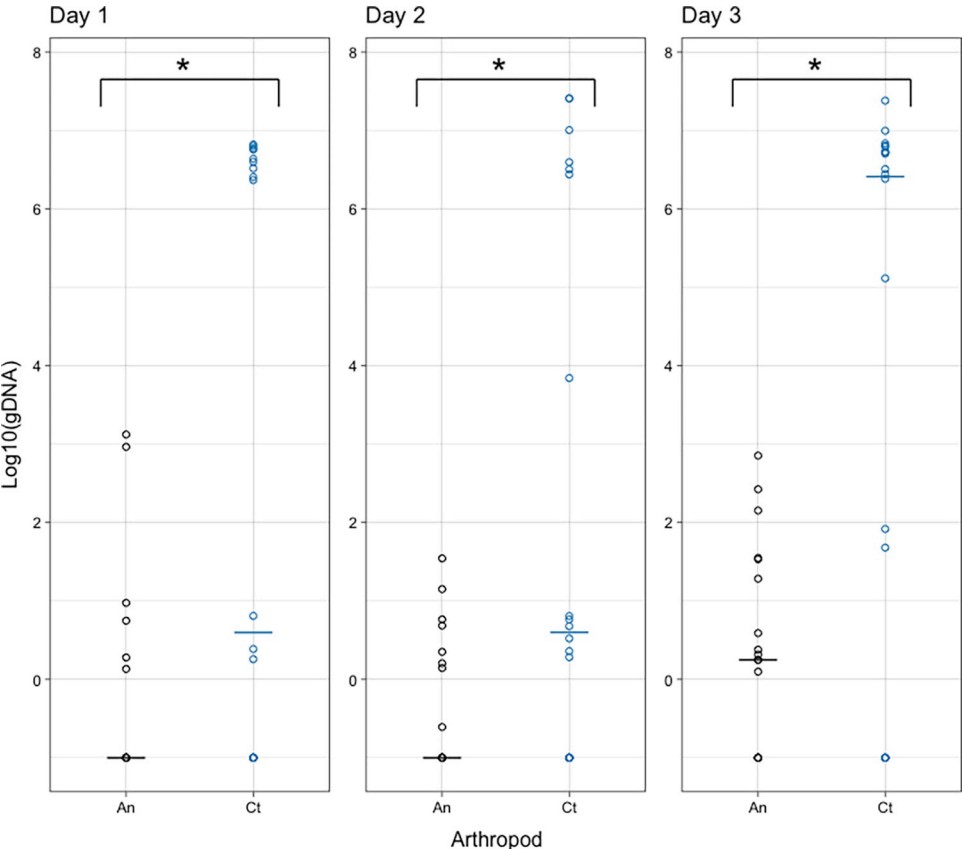

**Fig 2. Concentration (log10) of rickettsiae from individual mosquitoes (black circles) who were infected via cofeeding with cat fleas (blue circles) at 1, 2, or 3 days of cofeeding (lines represent medians).** There were significant differences in rickettsial load between the two arthropods (An = mosquitoes and Ct = cat fleas) at each timepoint, assessed by the Kruskal-Wallis test.

fleas, no statistical difference was observed in the proportion of *R. felis*-infected mosquitoes among the time points assessed. Comparing the proportions of infected between the two feeding arthropod populations, no significance difference was identified between the two vectors on any of the time points assessed. Combined, the data indicate that *An. quadrimaculatus* are susceptible to rickettsial infection via cofeeding with infected cat fleas.

Rickettsial load was compared among arthropod species and time points and no statistically significant difference across time for either mosquitoes or cat fleas was observed (**Fig 2**). However, when comparing cat fleas to mosquitoes at each day of cofeeding, donor cat fleas exposed to *R. felis* prior to cofeeding consistently had significantly higher rickettsial loads than the mosquitoes with which they were cofed (**Fig 2**).

## Discussion

The biology of *R. felis* is associated with varying degrees of disease, a distinct transmission cycle within the flea, and identification in a number of arthropods [1]. First coupled with human infection in the United States, *R. felis* and related RFLOs including *Rickettsia asembonensis* and *Candidatus* Rickettsia senegalensis have been associated with human disease worldwide [3]. Reported clinical signs in humans range from non-distinct flu-like symptoms to ulceration and more complex manifestations including neurological involvement [32–34].

Placed in the transitional group of *Rickettsia*, *R. felis* contains genetic and biological characteristics of both tick-associated spotted fever group (SFG) and insect-associated typhus group *Rickettsia* [35]. As obligate intracellular bacteria, arthropods often serve as both reservoirs and vectors of *Rickettsia* [36]. The most well-studied vector for *R. felis* is the common cat flea, which supports both vertical and horizontal transmission mechanisms [1]. In addition to cat fleas, *R. felis* has been described in a number of arthropods [4,37]. Furthermore, molecular detection of *R. felis* and RFLOs in field collected ticks and mosquitoes suggests that additional arthropod hosts, and possible transmission cycles, exist [11–15,17–19,38]. However, the route by which *R. felis* disseminates to infect such a large range of arthropod hosts is unknown.

Cofeeding transmission occurs when a pathogen is transmitted between two arthropods feeding in proximity on a vertebrate host, negating the intrinsic incubation period required for some pathogens during transmission events on a vertebrate host [39]. The cofeeding mechanism has been described for both flea- and tick- borne rickettsial pathogens, and occurs in the absence of a rickettsemic host [9,40]. Fleas deposit rickettsiae in the dermis at the vector-host skin interface resulting in dissemination across the skin [9], presumably where other arthropods, including ticks that have mouthparts in the same tissue, can acquire the agent during feeding events; thus, the transmission of *R. felis* from infected fleas to naïve ticks would be expected. In contrast, mosquitoes are capillary feeders and the mechanism by which rickettsiae transit between the dermis and associated capillaries targeted by mosquitoes remains to be examined. Although untested, the diversity of clinical symptoms associated with *R. felis*-rickettsioses, including skin presentation as well as systemic infection, may be related to the arthropod feeding duration (slow versus fast) or route of arthropod delivery (intradermal versus direct inoculation into the bloodstream). Further studies are needed to examine the potential for arthropod-dependent influence on rickettsial dissemination, elucidation of the host response to infection, and subsequent pathology in the vertebrate host.

The expansive distribution of *R. felis* is related to the cosmopolitan nature of cat fleas. There are several pieces of evidence supporting the potential for horizontal transmission of *R. felis* to vertebrate hosts, such as identification of *R. felis* in the salivary glands of cat fleas [41] and detection of *R. felis* in the blood of vertebrate hosts exposed to *R. felis*-infected fleas [5,42–44]. More recently, domestic canines were determined to be susceptible to *R. felis* infection and may serve as an infectious source of *R. felis* to arthropods other than fleas [45]. However, the mechanisms driving transmission of *R. felis* by arthropod vectors to vertebrate hosts or transmission between arthropod hosts are only in the initial stages of characterization. Intra- and inter-specific transmission of *R. felis* between cofeeding *R. felis*-infected cat fleas and naïve cat fleas and rat fleas via flea bite on vertebrate hosts has been demonstrated [9]. Similarly, ticks are known to transmit SFG *Rickettsia* via cofeeding [40]. The current study demonstrates cofeeding transmission by *R. felis*-infected donor fleas to naïve ticks. After only 24 hrs of cofeeding, uninfected recipient nymphal *D. variabilis* acquired *R. felis* with subsequent transstadial transmission to the adult stage. Under laboratory conditions, adult *D. variabilis* are susceptible to infection and subsequent transovarial transmission of *R. felis* for at least one generation [14]. However, the transovarial maintenance of *R. felis* infection in $F_1$ larvae was not assessed in the current study. Although not tested in the current study, the tick lifecycle stage and rickettsial load after exposure may contribute to the efficiency of transovarial transmission. As both cat fleas and ticks can be recovered from an individual vertebrate host, further studies are needed to assess the role of horizontal acquisition and vertical transmission routes in dissemination of *R. felis* in tick populations.

Mosquitoes are important vectors of pathogens that cause disease in humans and animals, but little is known about rickettsiae infection and transmission in mosquitoes. In the current study, *R. felis* acquired by *An. quadrimaculatus* can be maintained, although at a reduced

rickettsial load, within the mosquito host for up to 12 days. Twelve days is comparable to the time required for productive virus infection in mosquitoes after initial exposure [46,47]. Further study is required to examine the dissemination of *R. felis* to mosquito legs and saliva as an estimate of transmission potential. Laboratory and field studies postulate that mosquitoes may serve as vectors of *R. felis*. Indeed, there is evidence of transmission potential for *Anopheles* mosquitoes which in the laboratory are able to induce transient rickettsemia in mice [19]. Additionally, the detection of rickettsial DNA in field collected male and unfed females *Anopheles* mosquitoes suggests that vertical transmission may occur [17,18]. The detection of *R. felis* in molecular surveys of mosquitoes do not address the origin of rickettsial infection. Thus, the current study examined the potential for mosquitoes to acquire *R. felis* via cofeeding. Transmission of *R. felis* from infected cat fleas to naïve mosquitoes occurred after only 20 mins of cofeeding and acquisition efficiency increased corresponding to the duration infected fleas were fed on the vertebrate host. Additional studies, similar to ones demonstrating successive transmission from *R. felis*-infected fleas to uninfected fleas [9], are required to further characterize the role of mosquitoes in the ecology of *R. felis* distribution.

In summary, the current study demonstrated that *R. felis*-infected donor cat fleas can transmit the agent to naïve vectors via cofeeding on vertebrate hosts. The acquisition of rickettsiae by *D. variabilis* and *An. quadrimaculatus* suggests a new model for inter-class and inter-order transmission of *R. felis* possibly contributing to the wide-spread identification of this *Rickettsia* species in hematophagous arthropods. The arthropod species examined in this study were selected based on laboratory studies and species availability. Consistent with laboratory and field studies [17–19], the current results support the assertion that mosquitoes may have an active role in the ecoepidemiology of *R. felis* rickettsioses. The susceptibility of other species of ticks and mosquitoes is not known, but the diversity of arthropods associated with *R. felis* suggests widespread susceptibility of other species to *R. felis*. Additional studies are needed to elucidate the vector competency of arthropods other than fleas for *R. felis*, as defining potential alternative transmission mechanisms for rickettsial pathogens is necessary for designing sufficient intervention strategies to control the incidence of disease.

## Acknowledgments

We thank Dustin Miller (Centers for Disease Control and Prevention) and BEI Resources (NIAID, NIH) for consultation on mosquito use and biological resources, respectively.

## Author Contributions

**Conceptualization:** Chanida Fongsaran, Kevin R. Macaluso.

**Data curation:** Chanida Fongsaran, Krit Jirakanwisal, Natthida Tongluan, Allison Latour.

**Formal analysis:** Chanida Fongsaran, Rebecca C. Christofferson.

**Funding acquisition:** Kevin R. Macaluso.

**Investigation:** Chanida Fongsaran, Natthida Tongluan, Allison Latour.

**Methodology:** Chanida Fongsaran, Krit Jirakanwisal, Sean Healy.

**Project administration:** Kevin R. Macaluso.

**Resources:** Natthida Tongluan, Sean Healy.

**Software:** Rebecca C. Christofferson.

**Supervision:** Kevin R. Macaluso.

**Validation:** Chanida Fongsaran, Krit Jirakanwisal.

**Visualization:** Chanida Fongsaran, Rebecca C. Christofferson.

**Writing – original draft:** Chanida Fongsaran, Kevin R. Macaluso.

**Writing – review & editing:** Kevin R. Macaluso.

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
