## [Decision Letter · Decision Letter 0]

24 May 2022

Dear Dr. Macaluso,

Thank you very much for submitting your manuscript "The role of cofeeding arthropods in the transmission of an emerging rickettsial pathogen" for consideration at PLOS Neglected Tropical Diseases. As with all papers reviewed by the journal, your manuscript was reviewed by members of the editorial board and by several independent reviewers. The reviewers appreciated the attention to an important topic. Based on the reviews, we are likely to accept this manuscript for publication, providing that you modify the manuscript according to the review recommendations. 

Sincerely,

Job E Lopez, Ph.D.

Deputy Editor

Job Lopez

Deputy Editor

Reviewer's Responses to Questions

**Key Review Criteria Required for Acceptance?**

**Methods**

-Are the objectives of the study clearly articulated with a clear testable hypothesis stated?

-Is the study design appropriate to address the stated objectives?

-Is the population clearly described and appropriate for the hypothesis being tested?

-Is the sample size sufficient to ensure adequate power to address the hypothesis being tested?

-Were correct statistical analysis used to support conclusions?

-Are there concerns about ethical or regulatory requirements being met?

Reviewer #1: The method section was well written and described the methods used well.

Reviewer #2: I have read through the manuscript with these questions in mind. The objectives are clearly articulated and the study design was appropriate. The sample size and number of replicated experiments were appropriate to test the hypothesis and the statistical analyses were correct. I am fine with the overall design of the study.

Reviewer #3: I am happy with the well-described methodology in this article, Please refer to General Comments to the Authors.

**Results**

-Does the analysis presented match the analysis plan?

-Are the results clearly and completely presented?

-Are the figures (Tables, Images) of sufficient quality for clarity?

Reviewer #1: The results are clearly and completely presented including a table and figures.

Reviewer #2: The results from the study were clearly presented with the overall objectives in mind and the tables and figures were clear.

Reviewer #3: The results in this work are clearly presented, and I am happy with the quality of the Table and Figures in this manuscript.

**Conclusions**

-Are the conclusions supported by the data presented?

-Are the limitations of analysis clearly described?

-Do the authors discuss how these data can be helpful to advance our understanding of the topic under study?

-Is public health relevance addressed?

Reviewer #1: The conclusions are supported well be the data presented. In addition the authors discuss how the data presented can be helpful to advance our understanding of R. felis transmission.

Reviewer #2: The conclusions were supported by the data and the limitations were described. While the authors could work on clarifying the discussion in one of the main paragraphs, overall, this study clearly advances our understanding of the topic.

Reviewer #3: The conclusions of this manuscript are well supported by the performed assays. The work is highly related to the public health.

**Editorial and Data Presentation Modifications?**

Reviewer #1: Title: 

Would add to or replace emerging rickettsial pathogen the name of the pathogen Rickettsia felis.

Abstract:

Lines 44 and 48: Would suggest that you include the name of the vertebrate host e.g. rat.

Author summary:

Introduction:

Line 70: remove the word “by”. 

Line 78: suggest removing the word “vectors” and making the last word plural, i.e. arthropods.

Line 80: suggest replacing “vectors” with arthropods.

Materials and Methods:

Line 106: scapularis in italics i.e. scapularis-derived

Line 116: suggest defining qPCR as quantitative real-time PCR

Line 138: what was the portion of the donor cat fleas e.g. 10 or 10%?

Line 140: a portion, maybe this is written in the results?

Results:

Line 195: add Sprague Dawley or SD rat after vertebrate host.

Line 200: add number of positive adult ticks/number evaluated

Table 1:

Define ND

Discussion:

Line 252: Rickettsia senegalensis should be written as Candidatus Rickettsia senegalensis.

Reviewer #2: Line 43: transmit should be past-tense – transmitted

Line 45: There is no comma needed between observed and with.

Line 48: No comma needed between host and with

Line 53: The way this is written it seems to say that R felis is the common cat flea - take out the comma after felis

Line 60: Can take out 'other arthropods including' as ADT and mosquitoes were the only ones tested

Line 65: ‘with’ normally does not follow a semi-colon (;) so you can remove it. Might be better to phrase it: 'rash followed by complications involving neurological systems (1).'

Line 68: Better to change the tense: ‘have been increasing’ to ‘have increased’

Line 106: scapularis needs to be italicized

Lines 193-209: The data here is good but I struggled while I read with the question: how do they differentiate between live R. felis and DNA from dead R. felis in D variabilis nymphs? I realize the qPCR measures that - but I feel it could be stated more clearly in this section of the results so the reader isn't wondering what kind of DNA is being detected. For example, this aspect was taken care of in the mosquito section by presenting the linear changes in rickettsial concentration over the different time points.

Line 292: No need for semi-colon between ‘felis’ and ‘with’

Lines 301-321: There are several instances in this paragraph where there is a need to clarify ideas. Please re-work the whole paragraph so the ideas read more smoothly.

Lines 302-305: There are quite a few comma phrases here in this four line sentence. I would suggest making two sentences out of it so the ideas flow better.

Lines 308: no comma needed after mosquitoes. Could even change it to ‘which, in the lab, are able...’

Lines 309-311: sentence is a bit awkward - please smooth - two very different thoughts connected by an 'and'

Lines 311-315: Again, these are confusing sentences. Please work to smooth out the storyline as ideas feel like they're jumping around.

Lines 326-329: Key to highlight here is the need for transmission studies. I realize it is mentioned in the last sentence but it needs to be brought out more because only one aspect of competency has been demonstrated here (acquisition by the non-flea host) – transmission by the non-flea host still needs to be demonstrated.

Reviewer #3: I do not have advice to modify the existing data in this manuscript.

**Summary and General Comments**

Reviewer #1: Overall the manuscript is well and clearly written. The introduction of R. felis transmission by co-feeding with potential vectors/hosts will initiate further investigations to clarify the diversity and breath of vectors of this unusual rickettsial pathogen.

Reviewer #2: The nidality of arthropod-borne rickettsial disease systems are fascinating and, at times, can boggle the mind. The Rickettsia felis system is particularly interesting with publications constantly popping up, reporting its presence in all kinds of arthropods. This study takes it one step further and moves into the question ‘how does this rickettsial pathogen get around so freely?’ I really enjoyed reading this paper and commend the research team for thinking through the feeding trials in such a way that produced some pretty clear results. In summary, with some minor revisions and clarifications, I feel this paper will be ready - and will help to fill in some gaps in our knowledge regarding how R. felis moves between different hosts.

Reviewer #3: The project in this manuscript used several bioassays to explore the role of cofeeding arthropods in the transmission of Rickettsia felis, an important emerging rickettsial pathogen. This reviewer enjoyed reading this manuscript which is well-written. The methods related to this project are described in the details, and the conclusion of this manuscript is well supported by the performed assays. The authors are encouraged to address the below comments:

The authors may consider including the keyword- the organism (Rickettsia felis) as part of the manuscript title.

R. felis has been identified in multiple vectors, including different species of ticks and mosquitoes. The authors shall provide the rationale regarding the choice of the tick and mosquito species used in this project. In addition, please discuss if the conclusions drawn by the use of the tick and mosquito species in this work will apply well to other tick species and mosquito species.

The starting copy number of R. felis is very high ((3 × 10~10 rickettsiae/700 μl). Then, the detected rickessial copies in ticks and mosquitoes are much lower, particularly in mosquitoes. Could the R. felis positivity in mosquitoes be due to the carry-over of R. felis from the original organism? The qPCR, not the culture, was used to determine the copy number of R. felis, not the viability of the pathogen.

Regarding the transmission of R. felis from fleas to cofeeding mosquitoes, lines 48-50 reads: Similar to ticks, mosquitoes were able to acquire R. felis while cofeeding with infected cat fleas on a vertebrate host, with infection persisting in the mosquito for up to 3 dpe. In this experiment, the R. felis identified in mosquitoes can possibly acquire the organism from cat flea and/or the host? Or, more likely from the host?

PLOS authors have the option to publish the peer review history of their article (what does this mean?). If published, this will include your full peer review and any attached files.

Reviewer #1: No

Reviewer #2: No

Reviewer #3: Yes: Chengming Wang

Figure Files:

Data Requirements:

Reproducibility:

References

---

## [Editor Report · Decision Letter 1]

11 Jun 2022

Dear Dr. Macaluso,

We are pleased to inform you that your manuscript 'The role of cofeeding arthropods in the transmission of Rickettsia felis' has been provisionally accepted for publication in PLOS Neglected Tropical Diseases.

Best regards,

Job E Lopez, Ph.D.

Deputy Editor

Job Lopez

Deputy Editor

---

## [Editor Report · Acceptance letter]

21 Jun 2022

Dear Dr. Macaluso,

We are delighted to inform you that your manuscript, "The role of cofeeding arthropods in the transmission of *Rickettsia felis*," has been formally accepted for publication in PLOS Neglected Tropical Diseases.

Best regards,

Shaden Kamhawi

co-Editor-in-Chief

Paul Brindley

co-Editor-in-Chief
